# Propagation Constant Measurement Based on a Single Transmission Line Standard Using a Two-Port VNA

**DOI:** 10.3390/s23094548

**Published:** 2023-05-07

**Authors:** Ziad Hatab, Arezoo Abdi, Gregor Steinbauer, Michael Ernst Gadringer, Wolfgang Bösch

**Affiliations:** Christian Doppler Laboratory for Technology Guided Electronic Component Design and Characterization, Institute of Microwave and Photonic Engineering, Graz University of Technology, 8010 Graz, Austria

**Keywords:** microwave measurement, network analyzers, propagation constant, traceability

## Abstract

This study presents a new method for measuring the propagation constant of transmission lines using a single line standard and without prior calibration of a two-port vector network analyzer (VNA). The method provides accurate results by emulating multiple line standards of the multiline calibration method. Each line standard was realized by sweeping an unknown network along a transmission line. The network need not be symmetric or reciprocal, but must exhibit both transmission and reflection. We performed measurements using a slab coaxial airline and repeated the measurements on three different VNAs. The measured propagation constant of the slab coaxial airline from all VNAs was nearly identical. By avoiding disconnecting or moving the cables, the proposed method eliminates errors related to the repeatability of connectors, resulting in improved broadband traceability to SI units.

## 1. Introduction

The propagation constant is a critical parameter in transmission line analysis, providing valuable information about the electrical properties of materials at different frequencies. The need for accurate measurement of the propagation constant arises in various applications, such as material characterization [1,2,3,4] or the estimation of the characteristic impedance of transmission lines, which allows impedance renormalization in various vector network analyzer (VNA) calibration methods [5]. Furthermore, knowledge of the propagation constant allows the analysis of losses along a transmission line, which is a critical aspect in signal integrity applications [6,7]. In general, there are many reasons to measure the propagation constant of guided wave structures such as transmission lines.

There are several methods to measure the propagation constant using a two-port VNA, but the most-versatile method because of its broadband applicability is the multiline technique [8]. In this method, multiple lines of different lengths are measured to sample the traveling wave along the line standards in a broadband scheme. However, this approach has several drawbacks, including the need for multiple lines, the possibility of uncertainties in their geometry, and the requirement for accurate repeated connection or probing, all of which contribute to measurement uncertainties [9,10,11].

To address some of the problems of the multiline method, some techniques have been introduced, such as the multireflect method [12,13] and the line–network–network method [14,15,16,17]. The multireflect method uses multiple identical reflect standards with different offsets to provide broadband measurement of the propagation constant. However, because it requires multiple identical independent standards, it is susceptible to repeatability errors due to repeated connection or probing. In addition, the propagation constant must be solved using optimization techniques that could diverge if not well conditioned. The line–network–network method involves moving an unknown symmetric and reciprocal network along a transmission line and solving for the propagation constant using the derived similarity equations. This method has limitations, such as the restriction to three offsets, which limits the frequency range, and the requirement to use symmetric and reciprocal offset networks. The results of relative effective permittivity measurements using this method were presented in [18], highlighting the sensitivity and limitations of this solution.

It is noteworthy that there is a significant amount of literature discussing the broadband measurement of the propagation constant using only two line standards of varying lengths, commonly known as the line–line method [19,20,21,22]. Despite the different mathematical formulations used, all these methods are based on solving the characteristic polynomial of the eigenvalue problem associated with the thru–reflect–line (TRL) calibration [23]. However, because of the use of only two lines, which often have a significant length difference to cover lower frequencies, the result of the propagation constant exhibits multiple resonance peaks, caused by integer multiples of half-wavelength occurrences in the electrical length of the transmission lines. To mitigate this issue, some authors have proposed post-processing techniques to filter the resonance peaks [24,25].

There are several indirect techniques for determining the propagation constant of transmission lines, which involve evaluating the permittivity of materials separately. These methods can be broadly classified into two categories: the resonant method and the transmission/reflection method. The resonant method, described in [26,27], estimates the permittivity from the S-parameters at the resonant frequencies, resulting in measurements only at specific frequencies. In contrast, the transmission/reflection method estimates the permittivity of a sample placed between two waveguides from the measured transmission and reflection coefficients. This method can be implemented using various configurations such as free space, rectangular waveguide, and coaxial line, as discussed in [28,29,30,31,32].

This paper aimed to introduce a novel method for measuring the propagation constant using a single transmission line standard without the need for prior calibration of a two-port VNA. The proposed approach builds on the general idea presented in [14], where an unknown network is shifted along a transmission line. Unlike the previous solutions, our method is not limited by the number of offsets and can handle asymmetric and non-reciprocal networks. A weighted 4×4 eigenvalue problem is proposed to combine all offset measurements, inspired by the modified multiline method introduced in [33]. One of the key advantages of our approach is that it requires only one transmission line, which enables equations similar to those of the multiline method. The proposed method eliminates cable reconnection, ensuring high repeatability, with uncertainties mainly due to the dimensional motion of the unknown network and the intrinsic noise of the VNA. The effectiveness of the proposed method is demonstrated on a commercial slab coaxial airline with measurements conducted using three different VNA brands. This paper presents the mathematical derivation of the proposed method and experimental measurements, demonstrating its accuracy and high repeatability, even when different VNAs are used. The proposed method offers a promising alternative to existing methods for measuring the propagation constant.

The remainder of this paper is structured as follows. In Section 2, we provide a detailed explanation of the mathematical derivation of the eigenvalue problem formulation that allows for the adaptation of the multiline method. Subsequently, in Section 3, we discuss the use of normalized eigenvectors to extract the complex exponential terms, which contain the propagation constant, and the utilization of least squares to derive an accurate estimate of the propagation constant. In Section 4, we describe the experimental setup, where we perform measurements using various VNAs and present the measured propagation constant of the slab coaxial airline, as well as a comparison with EM simulation. Finally, a conclusion is given in Section 5.

## 2. Formulating the Eigenvalue Problem

The general idea of the measurement setup is to move an unknown network along a transmission line. For each movement of the network, either to the left or to the right, we created two offset elements that are complementary to each other. When the offset length is zero, the offset elements are reduced to a thru connection, which we refer to as the reference plane. An illustration of this concept is shown in Figure 1.

Before proceeding with the mathematical derivation, we need to define the sign convention for the offset shift. In our analysis, we define that moving the network to the right results in a positive offset, while moving the network to the left results in a negative offset. This convention is shown in Figure 2 as modeled by the error box model of a two-port VNA [34].

With the definition of the offsets in Figure 2, the measured T-parameters of the offset network by the length $l_{i}$ are given as follows:(1)$$M_{i}=\underset{k}{\underbrace{k_{a}k_{b}}}\underset{A}{\underbrace{\left[\begin{array}{cc} a_{11} & a_{12}\\ a_{21} & 1 \end{array}\right]}}L_{i}NL_{i}^{-1}\underset{B}{\underbrace{\left[\begin{array}{cc} b_{11} & b_{12}\\ b_{21} & 1 \end{array}\right]}},$$
where *k*, A, and B are the error terms of an uncalibrated two-port VNA. The matrices $L_{i}$ and N are given as follows:(2)$$L_{i}=\left[\begin{array}{cc} e^{-\gamma l_{i}} & 0\\ 0 & e^{\gamma l_{i}} \end{array}\right],\;N=\left[\begin{array}{cc} \frac{-S_{11}S_{22}+S_{12}S_{21}}{S_{21}} & \frac{S_{11}}{S_{21}}\\ \frac{-S_{22}}{S_{21}} & \frac{1}{S_{21}} \end{array}\right].$$

Here, γ represents the propagation constant of the transmission line and $\{S_{11},S_{12},S_{21},$$S_{22}\}$ are the S-parameters of the network N. The S-parameters of the offset network are generally unknown, and the network can be asymmetric or non-reciprocal. However, the network must satisfy some basic criteria, which are listed below:All S-parameters must be non-zero within the considered frequency range ($|S_{ij}|>0$).The S-parameters of the network should not change as the network is moved.The network should not lead to the generation of additional modes along the transmission line.

Although the first condition is unique to our method’s formulation, the remaining two conditions are also similar to the multiline method [8,33], which requires single-mode propagation and repeated error boxes. Fortunately, it is not difficult to design a system that satisfies these requirements. We will show this later in a Section 4, where we used a commercial sliding tuner that was not designed for our application, but met our conditions.

We now define the T-parameters of a new network by taking the difference in the T-parameters of two offset networks of different lengths $l_{i}$ and $l_{j}$, which is given by
(3)$$\begin{array}{cc} \;\bar{\;N\;}{\;}_{i,j} & =L_{i}NL_{i}^{-1}-L_{j}NL_{j}^{-1}\\ \; & =\nu_{i,j}\left[\begin{array}{cc} 0 & \frac{S_{11}}{S_{21}}e^{-\gamma (l_{i}+l_{j})}\\ \frac{S_{22}}{S_{21}}e^{\gamma (l_{i}+l_{j})} & 0 \end{array}\right], \end{array}$$
where
(4)$$\nu_{i,j}=e^{-\gamma (l_{i}-l_{j})}-e^{\gamma (l_{i}-l_{j})}.$$

The expression in (Equation 3) is very similar to a line standard in multiline calibration, but now, the line standard is described by an antidiagonal matrix and with additional multiplication factors. We define an equivalent measurement of a line standard by
(5)$$\;\bar{\;M\;}{\;}_{i,j}=M_{i}-M_{j}=kA\;\bar{\;N\;}{\;}_{i,j}B.$$

Similar to the multiline calibration, we also need an equation that describes the inverse of the measurements. This is given by
(6)$${\hat{\;\bar{\;M\;}\;}}_{i,j}=M_{i}^{-1}-M_{j}^{-1}=\frac{1}{k}B^{-1}{\hat{\;\bar{\;N\;}\;}}_{i,j}A^{-1},$$
where the matrix ${\hat{\;\bar{\;N\;}\;}}_{i,j}$ is given by
(7)$$\begin{array}{cc} {\hat{\;\bar{\;N\;}\;}}_{i,j} & =L_{i}^{-1}N^{-1}L_{i}-L_{j}^{-1}N^{-1}L_{j}\\ \; & =-\nu_{i,j}\left[\begin{array}{cc} 0 & \frac{S_{11}}{S_{12}}e^{-\gamma (l_{i}+l_{j})}\\ \frac{S_{22}}{S_{12}}e^{\gamma (l_{i}+l_{j})} & 0 \end{array}\right]. \end{array}$$

Given the expressions in (Equation 5) and (Equation 6), we can construct an eigenvalue problem in terms of A as follows:(8)$$\;\bar{\;M\;}{\;}_{i,j}{\hat{\;\bar{\;M\;}\;}}_{n,m}=A\;\bar{\;N\;}{\;}_{i,j}{\hat{\;\bar{\;N\;}\;}}_{n,m}A^{-1},$$
where the matrix product $\;\bar{\;N\;}{\;}_{i,j}{\hat{\;\bar{\;N\;}\;}}_{n,m}$ is given by
(9)$$\;\bar{\;N\;}{\;}_{i,j}{\hat{\;\bar{\;N\;}\;}}_{n,m}=-\kappa \nu_{i,j}\nu_{n,m}\left[\begin{array}{cc} e^{-\gamma (l_{i,j}^{+}-l_{n,m}^{+})} & 0\\ 0 & e^{\gamma (l_{i,j}^{+}-l_{n,m}^{+})} \end{array}\right],$$
with
(10)$$\kappa =\frac{S_{11}S_{22}}{S_{21}S_{12}},\;l_{i,j}^{+}=l_{i}+l_{j},\;l_{n,m}^{+}=l_{n}+l_{m}.$$

To have a valid eigenvalue problem, we need at least three unique offsets, where one of the offsets $l_{n}$ or $l_{m}$ can be equal to $l_{i}$ or $l_{j}$, but $l_{i}\neq l_{j}$, or vice versa. However, with three offsets, we have three possible pairs of the eigenvalue problems. In fact, for N≥3 offsets, we have $N\left(N-2\right)\left(N^{2}-1\right)/8$ possible pairs of eigenvalue problems. This is because, for a set of *N* offsets, we have N(N−1)/2 pairs, and when we create pairs from N(N−1)/2 pairs, we substitute the equation into itself, resulting in $N\left(N-2\right)\left(N^{2}-1\right)/8$ pairs of pairs.

To address the issue of multiple eigenvalue problems, we refer to our previous work in [33,35], where a similar problem was presented in the context of multiline calibration. This problem was solved by combining all measurements using a weighting matrix, reducing the problem to solving a single 4×4 eigenvalue problem, regardless of the number of lines. This method not only reduced the size of the problem, but also allowed us to express both error boxes A and B simultaneously in a single matrix using Kronecker product notation. By applying the techniques described in [33,35], we obtain the following set of equations:
(11a)$$\begin{array}{cc} \;\bar{\;M\;}\; & =kX\;\bar{\;N\;}\;, \end{array}$$
(11b)$$\begin{array}{cc} {\hat{\;\bar{\;M\;}\;}}^{T}P & =\frac{1}{k}{\hat{\;\bar{\;N\;}\;}}^{T}PX^{-1}, \end{array}$$
with
(12a)$$\begin{array}{cc} X & =B^{T}⊗A, \end{array}$$
(12b)$$\begin{array}{cc} \;\bar{\;M\;}\; & =\left[\begin{array}{ccc} \mathrm{vec}\left(\;\bar{\;M\;}{\;}_{1,2}\right) & \cdots & \mathrm{vec}\left(\;\bar{\;M\;}{\;}_{i,j}\right) \end{array}\right], \end{array}$$
(12c)$$\begin{array}{cc} \;\bar{\;N\;}\; & =\left[\begin{array}{ccc} \mathrm{vec}\left(\;\bar{\;N\;}{\;}_{1,2}\right) & \cdots & \mathrm{vec}\left(\;\bar{\;N\;}{\;}_{i,j}\right) \end{array}\right], \end{array}$$
(12d)$$\begin{array}{cc} \hat{\;\bar{\;M\;}\;} & =\left[\begin{array}{ccc} \mathrm{vec}\left({\hat{\;\bar{\;M\;}\;}}_{1,2}\right) & \cdots & \mathrm{vec}\left({\hat{\;\bar{\;M\;}\;}}_{i,j}\right) \end{array}\right], \end{array}$$
(12e)$$\begin{array}{cc} \hat{\;\bar{\;N\;}\;} & =\left[\begin{array}{ccc} \mathrm{vec}\left({\hat{\;\bar{\;N\;}\;}}_{1,2}\right) & \cdots & \mathrm{vec}\left({\hat{\;\bar{\;N\;}\;}}_{i,j}\right) \end{array}\right], \end{array}$$
(12f)$$\begin{array}{cc} P & =\left[\begin{array}{cccc} 1 & 0 & 0 & 0\\ 0 & 0 & 1 & 0\\ 0 & 1 & 0 & 0\\ 0 & 0 & 0 & 1 \end{array}\right],\;\mathrm{where},\;P=P^{-1}=P^{T}. \end{array}$$

The details of the definition and properties of the Kronecker product (⊗) and the matrix vectorization ($\mathrm{vec}\left(\right)$) can be found in [36].

We now formulate the main eigenvalue problem by defining a new matrix W, which we multiply on the right side of ([Disp-formula FD11a-sensors-23-04548]). We call this matrix the weighting matrix. In the next step, we constructed the weighted eigenvalue problem by multiplying the new equation on the left side of ([Disp-formula FD11b-sensors-23-04548]). This results in
(13)$$\underset{F}{\underbrace{\;\bar{\;M\;}\;W{\hat{\;\bar{\;M\;}\;}}^{T}P}}=X\underset{H}{\underbrace{\;\bar{\;N\;}\;W{\hat{\;\bar{\;N\;}\;}}^{T}P}}X^{-1}.$$

The expression presented in (Equation 13) represents a similarity problem between the matrices F and H, with X as the transformation matrix. The purpose of introducing the weighting matrix W is to transform this similarity problem into an eigenvalue problem by forcing H into a diagonal form. It turns out that if W is any non-zero skew-symmetric matrix, then H takes a diagonal form [33]. However, we do not only want to diagonalize H, but also want to maximize the distance between the eigenvalues, which in turn minimizes the sensitivity in the eigenvectors [37]. For multiline calibration, the optimal form of W was derived in [33], and since the formulation in (Equation 13) is similar to that discussed in [33], we used the same choice of W with some scaling modifications. The optimal weighting matrix W can be written as follows, taking into account the scaling factors:(14)$$W^{H}=-\kappa \left(zy^{T}-yz^{T}\right),$$
where
(15a)$$\begin{array}{cc} y^{T} & =\left[\begin{array}{ccc} \nu_{1,2}e^{\gamma l_{1,2}^{+}} & \ldots & \nu_{i,j}e^{\gamma l_{i,j}^{+}} \end{array}\right], \end{array}$$
(15b)$$\begin{array}{cc} z^{T} & =\left[\begin{array}{ccc} \nu_{1,2}e^{-\gamma l_{1,2}^{+}} & \ldots & \nu_{i,j}e^{-\gamma l_{i,j}^{+}} \end{array}\right]. \end{array}$$

As a result of choosing W as defined in (Equation 14), the expression in (Equation 13) takes an eigendecomposition form, as given below.
(16)$$F=X\left[\begin{array}{cccc} 0 & 0 & 0 & 0\\ 0 & \lambda & 0 & 0\\ 0 & 0 & -\lambda & 0\\ 0 & 0 & 0 & 0 \end{array}\right]X^{-1},$$
where λ is real-valued and proportional to the square Frobenius norm of the matrix W, given by
(17)$$\lambda =\frac{1}{2}{∥W∥}_{F}^{2}=\frac{1}{2}\sum_{i,j}{\left|w_{i,j}\right|}^{2}.$$

There are two ways to compute W: The first is the direct method, where we already know the propagation constant γ and the factor κ, which describes the unknown network. Naturally, the first option is not practical since both γ and κ are unknown. The better option is to apply a rank-2 Takagi decomposition to the left side of the following equation, as described in [35] for multiline calibration.
(18)$$\underset{\mathrm{measurement}}{\underbrace{{\hat{\;\bar{\;M\;}\;}}^{T}P\;\bar{\;M\;}\;}}=\underset{\mathrm{model}}{\underbrace{{\hat{\;\bar{\;N\;}\;}}^{T}P\;\bar{\;N\;}\;}}.$$

Note that the left side of (Equation 18) contains only the measurement data, while the right side describes the model. Furthermore, the error boxes are not present in (Equation 18). To determine W, we need to calculate the rank-2 Takagi decomposition. This was performed in two steps. First, we computed the rank-2 of (Equation 18) via singular-value decomposition (SVD), and then, we applied the Takagi decomposition to decompose the matrix into its symmetric basis [38]. This looks as follows:(19)$${\hat{\bar{N}}}^{T}P\bar{N}=\underset{\mathrm{rank}-2\;\mathrm{SVD}\;\mathrm{from}\;\mathrm{measurement}}{\underbrace{s_{1}u_{1}v_{1}^{H}+s_{2}u_{2}v_{2}^{H}}}=\underset{\mathrm{Takagi}}{\underbrace{GG^{T}}}$$

Then, the weighting matrix is given by
(20)$$W^{H}=\pm G\left[\begin{array}{cc} 0 & j\\ -j & 0 \end{array}\right]G^{T}$$

The derivation process of the matrix W is described in more detail in [35]. To resolve the sign ambiguity, one approach is to select the answer that has the smallest Euclidean distance to a known estimate. Such an estimate can be obtained from approximate knowledge of the material properties of the transmission line.

The last step is the solution of the eigenvectors described by X in (Equation 16). The solution of the eigenvectors was discussed in [33]. It is worth noting that we cannot solve the matrix X uniquely, but only up to a diagonal matrix multiplication. Therefore, to define a unique solution for X, we normalized its columns so that the diagonal elements are equal to one. This is written as follows:(21)$$\tilde{X}=X\mathrm{diag}{\left(a_{11}b_{11},b_{11},a_{11},1\right)}^{-1},$$
where $a_{11}$ and $b_{11}$ are part of the error boxes A and B (see (Equation 1) and ([Disp-formula FD12a-sensors-23-04548])).

## 3. Least-Squares Solution for the Propagation Constant

Knowing $\tilde{X}$ from the eigenvector solutions, we can extract the complex exponential terms that contain the propagation constant. To do this, we first multiplied the inverse of the normalized error terms to all vectorized measurements of the offset network. This is given by
(22)$$E={\tilde{X}}^{-1}M=\mathrm{diag}\left(ka_{11}b_{11},kb_{11},ka_{11},1\right)N^{\prime },$$
where
(23a)$$\begin{array}{cc} M & =\left[\begin{array}{ccc} \mathrm{vec}\left(M_{1}\right) & \cdots & \mathrm{vec}\left(M_{N}\right) \end{array}\right], \end{array}$$
(23b)$$\begin{array}{cc} N^{\prime } & =\left[\begin{array}{ccc} \mathrm{vec}\left(L_{1}NL_{1}^{-1}\right) & \cdots & \mathrm{vec}\left(L_{N}NL_{N}^{-1}\right) \end{array}\right]. \end{array}$$

Since we do not know the remaining error terms $k,a_{11},b_{11}$, as well as the S-parameters of the network N, we need to choose a reference offset to eliminate these unknown factors. For simplicity, we chose the first offset, which we define as zero, i.e., $l_{1}=0$ (any other choice is also valid). As a result, the positive and negative complex exponential terms are given as follows, using the indexing notation based on Python.
(24a)$$\begin{array}{cc} E[1,1:]/E[1,0]= & \left[\begin{array}{cccc} e^{2\gamma l_{2}} & e^{2\gamma l_{3}} & \ldots & e^{2\gamma l_{N}} \end{array}\right], \end{array}$$
(24b)$$\begin{array}{cc} E[2,1:]/E[2,0]= & \left[\begin{array}{cccc} e^{-2\gamma l_{2}} & e^{-2\gamma l_{3}} & \ldots & e^{-2\gamma l_{N}} \end{array}\right]. \end{array}$$

Now that we have the complex exponential terms, we can extract the exponents using the complex logarithm function and determine γ using the least-squares method, while taking care of any phase unwrapping. First, since we have both the positive and negative complex exponential terms, we can account for both by averaging them. This was performed by defining a new vector τ:(25)$$\tau ={\left[\begin{array}{ccc} \frac{e^{2\gamma l_{2}}+1/e^{-2\gamma l_{2}}}{2} & \cdots & \frac{e^{2\gamma l_{N}}+1/e^{-2\gamma l_{N}}}{2} \end{array}\right]}^{T}.$$

The next step is to calculate the logarithm to extract the exponents, which is given by
(26)$$\phi =\log\left(\tau \right)+j2\pi n,\;\mathrm{where},\;n\in Z^{N-1}.$$

The phase unwrapping factor n can be estimated by rounding the difference between ϕ and an estimated value. This is given by
(27)n=roundImϕ−2γestl2π,
where $\gamma_{\mathrm{est}}$ is a known approximation for γ and l is a vector containing all length offsets except the reference zero offset. The initial estimate for $\gamma_{\mathrm{est}}$ can be derived from the material properties of the transmission line.

Finally, we can determine γ through the weighted least-squares [39]:(28)$$\gamma =\frac{l^{T}V^{-1}\phi }{l^{T}V^{-1}l},$$
where $V^{-1}$ is given by
(29)$$V^{-1}=I_{\left(N-1)\times (N-1\right)}-\frac{1}{N}1_{N-1}1_{N-1}^{T},$$

The matrix I is the identity matrix, and 1 is a vector of ones. The weighting matrix $V^{-1}$ is necessary because each measurement has a common reference, which is $l_{1}$. Therefore, the correlation between the measurements had to be taken into account by the matrix $V^{-1}$ [39].

Figure 3 summarizes the mathematical derivation presented in this and the previous sections and provides a visual representation of the steps taken to compute the propagation constant.

## 4. Experiment and Discussion

### 4.1. Measurement Setup

For demonstration purposes, we used the slide screw tuner 8045P from Maury Microwave as an implementation of the offset network, where the transmission line was a slab coaxial airline that supported frequencies up to 18 GHz. The tuner is depicted in Figure 4.

For our method to work, we required that the unknown network (i.e., the tuner element) be both reflective and transmissive, as the factor κ in (Equation 9) can explode to infinity if the network is only reflective and can be zero if the network is only transmissive. Ideally, we wanted κ=1 to minimize its effect on the eigenvalue problem. However, we also wanted to avoid scenarios where the network causes the generation of additional modes or resonances. Therefore, we adjusted the tuner with an already calibrated VNA to tune the network to a desired response, as shown in Figure 5. It should be noted that this step of tuning the tuner with an existing calibrated VNA was only necessary because the tuner was a commercial product designed for circuit matching applications and not for our purposes. If we were designing the network ourselves, we would not need to measure it with a calibrated VNA because we would have already designed it to meet our frequency specifications. Furthermore, the S-parameters of the network were never explicitly used in the derivation of the propagation constant.

As shown in Figure 5, we set the lower frequency to 3 GHz to avoid very low return loss and resonances. We then measured the airline using different uncalibrated VNAs. This was performed to demonstrate that, even if we changed the measurement system, we would still obtain consistent results because the error boxes would not be affected by uncertainties caused by connector and cable movement. For the offset lengths, we chose {0,21,66,81,84,93,117,123,171,192}mm, which ensured that the eigenvalue λ in (Equation 16) does not reach zero in the target frequency range.

The VNAs used for the measurements were: Anritsu VectorStar, R&S ZNA, and Keysight ENA. The ENA is limited to 14 GHz. All VNAs were placed in the same room to provide the same room conditions. The power level and IF bandwidth for all VNAs were set to 0 dBm and 100 Hz, respectively. Due to the low loss of the airline, an average measurement of 50 frequency sweeps was calculated to reduce noise. Pictures of the three instruments are shown in Figure 6.

### 4.2. Results and Discussion

All measurements of the different offsets were taken without prior calibration of the VNAs. The collected data were then read in Python using the *scikit-rf* package [40]. In Figure 7, we show the measured magnitude response of $S_{11}$ and $S_{21}$ from all three VNAs for the offset 123mm. From the figure, we can see that all three VNAs give different responses because the error boxes are different for each VNA.

After collecting all raw measurements for all the offsets and from all the VNAs, the data were processed to extract the propagation constant according to the discussion in Section 2 and Section 3. For an easier and better interpretation of the extracted propagation constant, we plotted in Figure 8 the real part of the relative effective permittivity and the loss per unit length of the slab coaxial airline from all three VNA measurements. The real part of the relative effective permittivity and the loss per unit length are calculated from the propagation constant as follows:(30)ϵr,eff′=−Rec0γ2πf2(Unitless),loss=20×10−2ln10Reγ(dB/cm),
where $c_{0}$ is the speed of light in vacuum and *f* is the frequency.

The relative effective permittivity and loss per unit length results presented in Figure 8 showed clear agreement between all VNA measurements, demonstrating the high repeatability of the proposed method even when using different VNA setups.

We also performed an EM simulation with the dimensional parameters of the airline given in Figure 4. Unfortunately, we did not have information on the metal types of the inner and outer conductors. From the appearance of the inner conductor, we believe it was made of some kind of brass. For the ground plates, we believe they were made of aluminum because they had a black anodized coating, which is typical for aluminum components. The anodized layer is often based on aluminum oxide and typically has a relative permittivity of 8.3 [41]. Since the thickness of the oxide layer and the exact conductivity of brass were unknown, we ran some values for the thickness of the anodic layer and the conductivity of brass. We found that a coating thickness of 15μm and a relative conductivity of 35% International Annealed Copper Standard (IACS) overlapped with the measurement shown in Figure 8. The value obtained for the thickness of the anodic layer is quite typical to obtain a dark black coating [42]. The conductivity of the inner conductor of 35% IACS (=20.3 MS/m) was within the range of common brass types [43].

The purpose of the simulation was to show that the results obtained from the proposed method of measuring the propagation constant do indeed translate into the realistic properties of the transmission line. In fact, with the proposed method, one could characterize materials in reverse, as in our case the conductivity of the metal.

Another aspect that may be of interest is the quality of the extracted propagation constant by varying the length and number of offsets. In the results shown in Figure 8, we used 10 offsets ranging from 0 to 192 mm. Now, we consider different cases. These cases are listed in Table 1.

In Figure 9, we show the results of the relative effective permittivity and the loss per unit length of the slab coaxial airline from the VectorStar VNA measurements for all the cases mentioned in Table 1. Cases 1 and 2 show the results when only three offsets were considered. Case 2 differs from Case 1 in that we replaced the last offset with a much longer offset. The results of both Cases 1 and 2 were poor and showed multiple resonances. For Case 2, we saw more resonances than for Case 1. This was the result of the eigenvalue crossing zero at multiple frequencies (see Figure 10). In Case 3, we spread the offsets further to include five offsets. We can see a clear improvement over Cases 1 and 2. We could further improve the accuracy of the extracted relative effective permittivity and loss per unit length by further spreading the offsets as in Case 4, where we used seven offsets. In Case 4, we obtained results of similar accuracy to the case of using all 10 offset lengths.

The quality of the extracted propagation constant depends on the eigenvalue λ as defined in (Equation 16). As the eigenvalue approached zero, the eigenvectors became more sensitive, which in turn affects the calculation of the extracted propagation constant. To visualize the differences between different scenarios, we present a scaled representation of the eigenvalue λ for each case. This scaled representation excludes the influence of the network through the common factor κ, which was invariant over all offset lengths. Since κ>0 was established earlier, variations in the eigenvalues can only be induced by the choice of offset lengths. Accordingly, we define the normalized version of the eigenvalue by dividing it by the absolute value of κ, as shown below:(31)$$\lambda =\frac{1}{2}{∥W∥}_{F}^{2}=\frac{{\left|\kappa \right|}^{2}}{2}{∥zy^{T}-yz^{T}∥}_{F}^{2}\;⟹\;\lambda^{\prime }=\frac{\lambda }{{\left|\kappa \right|}^{2}}=\frac{1}{2}{∥zy^{T}-yz^{T}∥}_{F}^{2}$$

In Figure 10, we present a plot of the scaled eigenvalue normalized to its maximum value, which facilitates a consistent comparison as the number of offsets varies. As illustrated in the figure, for Cases 1 and 2, the eigenvalue exhibited multiple zero crossings at various frequencies. Similarly, in Case 3, the eigenvalue approached zero at several instances, although to a lesser extent than in Cases 1 and 2. In contrast, in Case 4, the eigenvalue never reached zero, but attained values closer to zero at specific frequencies than when all 10 offsets were utilized. Ideally, a flat eigenvalue over frequency would be preferred, but this would necessitate employing even more offsets. This is not different from the multiline calibration approach proposed in [33], where a finer spacing between lines resulted in a flatter eigenvalue over the frequency. Therefore, utilizing a broader range of offset lengths was highly advantageous for enhancing the accuracy of the results across the frequency. It is also noteworthy that the eigenvalue possessed a bandpass characteristic, whereby the lowest and highest frequency limits were bound by the largest and smallest relative offset, respectively.

For comparison, it is worth noting that the multiline method necessitates the measurement of multiple line standards of different lengths, a process that can introduce errors due to connector repeatability. Achieving high repeatability in this context poses a significant mechanical challenge, especially concerning connectors, and automating this process represents an even greater hurdle. In contrast, our proposed method eliminates the need for physical contact between the sliding element and the transmission line. Furthermore, although the sliding process was performed manually in the example presented, it could be automated by employing a linear actuator, thus eliminating the need for any user interaction with the measurement system. In Table 2, we summarize the comparison between the proposed method and existing works in measuring the propagation constant of transmission lines.

## 5. Conclusions

We presented a new broadband method for measuring the propagation constant of transmission lines that does not require the prior calibration of a two-port VNA or the use of multiple line standards. This method provides accurate results by emulating the use of multiple line standards through sweeping an unknown network along a transmission line. The shifted network does not have to be symmetric or reciprocal, but it must exhibit both transmission and reflection properties and remain invariant when moved along the line. The experimental results obtained using different VNAs on a slab coaxial airline with a slider tuner showed consistent agreement with each other and with the EM simulation.

One of the significant advantages of the proposed method is that it uses the same eigenvalue formulation as multiline calibration, but without the need for disconnecting or moving the cables. As a result, it eliminates errors related to connector repeatability and provides improved broadband traceability to the SI units. Moreover, since the offsets are implemented by simply moving the unknown network laterally, the process can be easily automated using an automated linear actuator. Therefore, the proposed method can accurately measure the propagation constant without requiring any physical interaction from the user on the measurement system.

## Figures and Tables

**Figure 1 sensors-23-04548-f001:**
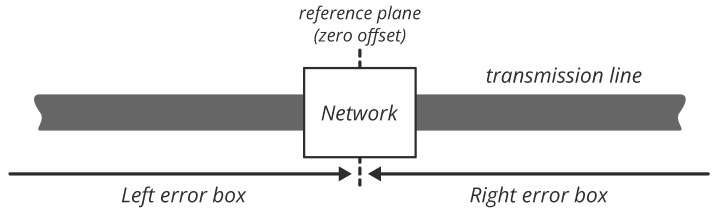
Illustration of network offset on a transmission line.

**Figure 2 sensors-23-04548-f002:**
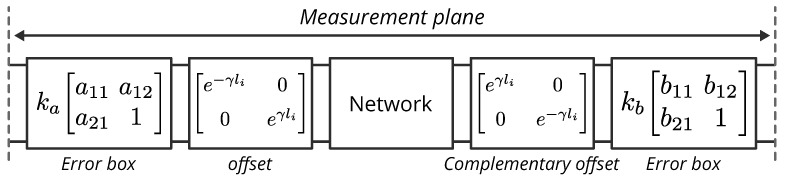
VNA two-port error box model of a network offset by a length $l_{i}$ (positive or negative). Each offset results in two complementary offset boxes. All blocks are given by their T-parameters.

**Figure 3 sensors-23-04548-f003:**
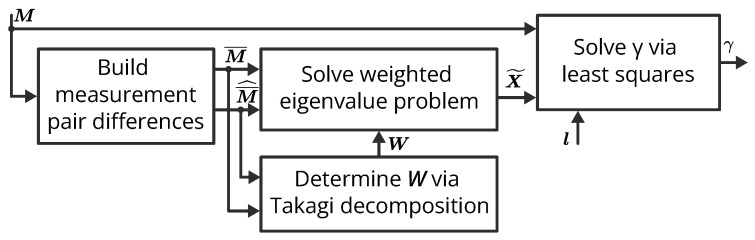
Block diagram summary of the proposed propagation constant measurement method. The matrix M contains the T-parameter measurements of all offsets. The vector l contains the relative length of the offsets with respect to the reference offset (i.e., the zero offset).

**Figure 4 sensors-23-04548-f004:**
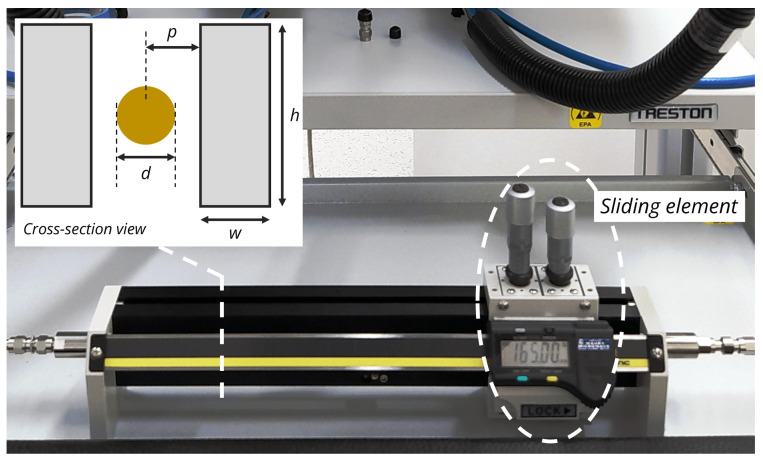
Maury Microwave 8045P tuner. The cross-section dimensions of the airline are given as follows: w=9.398mm, h=40.691mm, d=3.040mm, and p=2.778mm.

**Figure 5 sensors-23-04548-f005:**
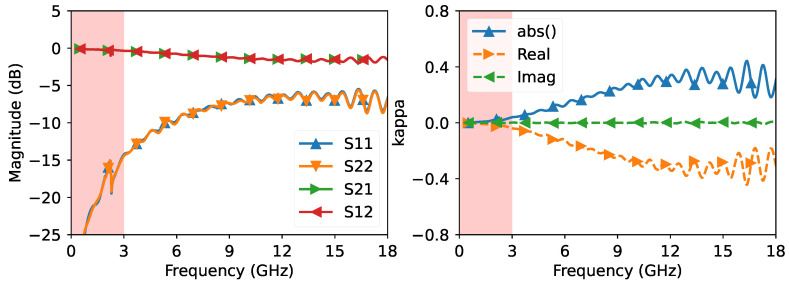
Calibrated measurement of the tuner after tuning. The highlighted frequency range below 3 GHz is not usable due to small reflection and resonance.

**Figure 6 sensors-23-04548-f006:**
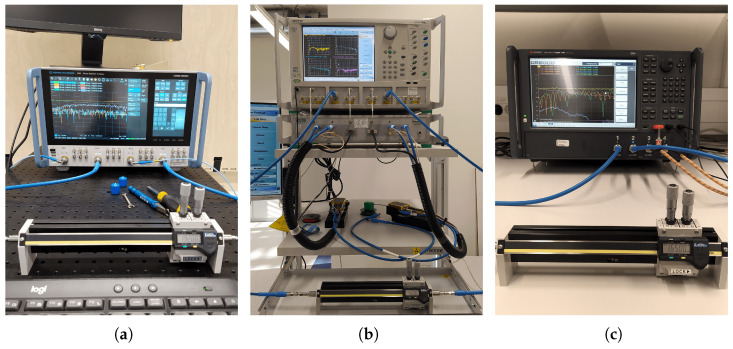
The VNAs used for the measurements. (**a**) Rohde & Schwarz ZNA, (**b**) Anritsu VectorStar, and (**c**) Keysight ENA.

**Figure 7 sensors-23-04548-f007:**
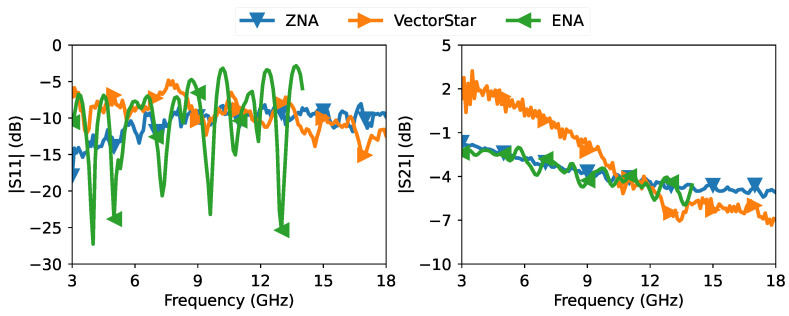
Raw measurements from the three VNAs of the magnitude response of $S_{11}$ and $S_{21}$ of the 8045P tuner, at an offset location of 123mm.

**Figure 8 sensors-23-04548-f008:**
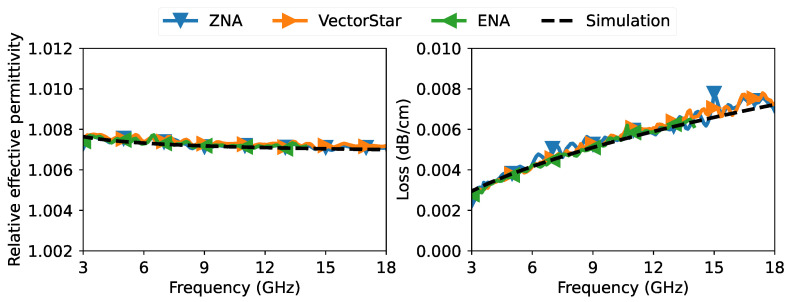
Extracted measurement of relative effective permittivity and loss per unit length of the slab coaxial airline, as well as an EM-simulated results for an anodic coating of 15 μm and an inner conductor conductivity of 35% IACS.

**Figure 9 sensors-23-04548-f009:**
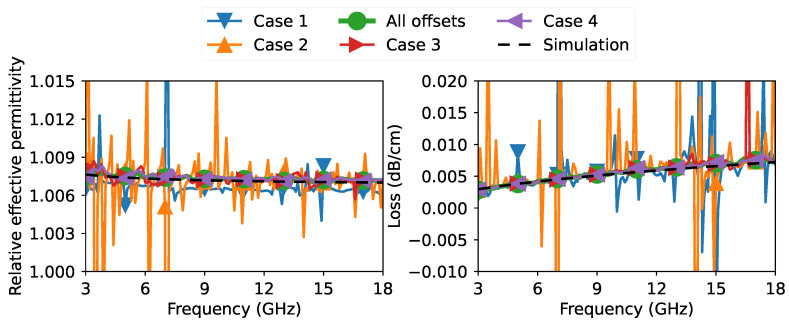
Extracted measurement of relative effective permittivity and loss per unit length of the slab coaxial airline for various combinations of offsets from the VectorStar VNA measurements.

**Figure 10 sensors-23-04548-f010:**
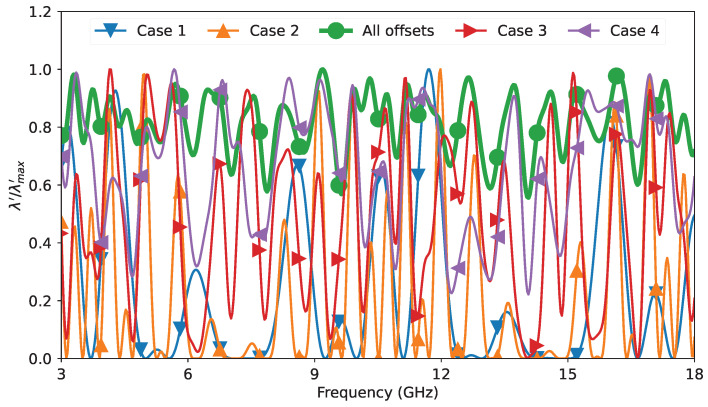
The scaled eigenvalue normalized to its maximum for the investigated offset length cases.

**Table 1 sensors-23-04548-t001:** Considered cases of different offset lengths.

Cases	Offset Lengths (mm)
Case 1	0,21,81
Case 2	0,21,192
Case 3	0,21,66,117,192
Case 4	0,21,81,93,117,123,192
All offsets	0,21,66,81,84,93,117,123,171,192

**Table 2 sensors-23-04548-t002:** Comparison between the proposed method against previous published work on extracting the propagation constant of transmission lines.

Method	Frequency Range	Solution Method	Measured Standards	Repeatability Accuracy
Multinetwork (this paper).	Broadband.	Matrix decomposition and linear least squares.	Single line with sweepable network.	Very high.
Multiline [8,33,39].	Broadband.	Matrix decomposition and linear least squares.	Multiple lines of different lengths.	Good.
Two lines [19,20,21,22,23].	Limited.	Quadratic equation.	Two lines of different lengths.	High.
LNN method [14,15,16,17,18].	Limited.	Quadratic equation.	Single line with sweepable symmetric and reciprocal network.	Good.
Multireflect [12,13].	Broadband.	Non-linear optimization.	Multiple symmetric reflect at different offsets.	Good.

## Data Availability

Publicly available datasets were analyzed in this study. These data can be found here: https://github.com/ZiadHatab/two-port-single-line-propagation-constant (accessed on 27 April 2023).

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
