# Peer review of "Propagation Constant Measurement Based on a Single Transmission Line Standard Using a Two-Port VNA"

_sensors, 2023, doi:10.3390/s23094548_

Round 1
Reviewer 1 Report
Minor modifications required are focused on two aspects:
1) State-of-the-Art in the introduction has been well established, except that two equivalent methods in free space are missing: Fenner method (Fenner, R. A., Rothwell, E. J., & Frasch, L. L. (2012). A comprehensive analysis of free-space and guided-wave techniques for extracting the permeability and permittivity of materials using reflection-only measurements. Radio Science. https://doi.org/10.1029/2011RS004755) , and Pouliguen method (Pometcu, L., Sharaiha, A., Benzerga, R., Tamas, R. D., & Pouliguen, P. (2016). Method for material characterization in a non-anechoic environment. Applied Physics Letters, 108(16). https://doi.org/10.1063/1.4947100)
2) From measurements, the calculations are very complex and exhaustive for permittivity extraction compared to others methods. The only applicative example has been made on air permittivity extraction which is not demonstrative. It should be more convincing on others materials with relative permittivity different from unity. A measurement on another material than air is mandatory for Technique validation.
Author Response
Thank you for reviewing our paper. Below you find our response to your feedback.
- “State-of-the-Art in the introduction has been well established, except that two equivalent methods in free space are missing: Fenner method (Fenner, R. A., Rothwell, E. J., & Frasch, L. L. (2012). A comprehensive analysis of free-space and guided-wave techniques for extracting the permeability and permittivity of materials using reflection-only measurements. Radio Science. https://doi.org/10.1029/2011RS004755) , and Pouliguen method (Pometcu, L., Sharaiha, A., Benzerga, R., Tamas, R. D., & Pouliguen, P. (2016). Method for material characterization in a non-anechoic environment. Applied Physics Letters, 108(16). https://doi.org/10.1063/1.4947100).”
Thank you for your remark. The references have been mentioned in the introduction section (see refs. [31] and [32]).
- Fenner, R.A.; Rothwell, E.J.; Frasch, L.L. A comprehensive analysis of free-space and guided-wave techniques for extracting the 438 permeability and permittivity of materials using reflection-only measurements. Radio Science 2012, 47. https://doi.org/10.1029/ 439 2011RS004755. 440
- Pometcu, L.; Sharaiha, A.; Benzerga, R.; Tamas, R.D.; Pouliguen, P. Method for material characterization in a non-anechoic 441 environment. Applied Physics Letters 2016, 108, 161604. https://doi.org/10.1063/1.4947100.
- “From measurements, the calculations are very complex and exhaustive for permittivity extraction compared to others methods. The only applicative example has been made on air permittivity extraction which is not demonstrative. It should be more convincing on others materials with relative permittivity different from unity. A measurement on another material than air is mandatory for Technique validation.”
Thank you for your comment. The mathematics may appear complicated, but for practical applications, only four steps are required to obtain the results, as we have summarized in Figure 3. Additionally, methods like the free-space and waveguide approach require prior calibration of the VNA, which is part of the math. However, our solution does not require any prior calibration. In fact, you can use the algorithm directly in Python without going through the math, as we have already made the code publicly available on GitHub: https://github.com/ZiadHatab/two-port-single-line-propagation-constant.
Regarding only measuring air, this is because the Tuner 8045P is the only device we currently have that has a sliding tool. Therefore, we would need to custom design and manufacture our sliding device to measure other materials. Currently, we are limited to the presented approach. Concurrently, we are looking into ways to easily manufacture the sliding mechanism for measuring transmission lines with different materials.
Reviewer 2 Report
A new method for measuring the propagation constant of transmission lines was proposed in this paper. The theoretical background and experimental measurement methods of this method are described in detail. This article is well written and innovative. I suggest that this article be published after a minor revision, with the following suggestions:
1. The objective of this article should probably be described more clearly in the introduction section.
2. Figure 5, the authors stated that "The highlighted frequency range below
3GHz is not usable due to small reflection and resonance". Do you have any way to solve this problem?
3. Section 4.2, could you make a table that compares the advantages and disadvantages of existing propagation constant measurement methods?
Author Response
Thank you for reviewing our paper. Below you find our response to your feedback.
- “The objective of this article should probably be described more clearly in the introduction section.”
Thank you for your comment. We edited the full paragraph discussing our method in the introduction section.
This paper aims to introduce a novel method for measuring the propagation constant using a single transmission line standard without the need for prior calibration of a two-port VNA. The proposed approach builds on the general idea presented in [14], where an unknown network is shifted along a transmission line. Unlike the previous solutions, our method is not limited by the number of offsets and can handle asymmetric and non-reciprocal networks. A weighted 4 × 4 eigenvalue problem is proposed to combine all offset measurements, inspired by the modified multiline method introduced in [33]. One of the key advantages of our approach is that it requires only one transmission line, which enables equations similar to those of the multiline method. The proposed method eliminates cable reconnection, ensuring high repeatability, with uncertainties mainly due to the dimensional motion of the unknown network and intrinsic noise of the VNA. The effectiveness of the proposed method is demonstrated on a commercial slab coaxial airline with measurements conducted using three different VNA brands. This paper presents the mathematical derivation of the proposed method and experimental measurements, demonstrating its accuracy and high repeatability, even when different VNAs are used. The proposed method offers a promising alternative to existing methods for measuring the propagation constant. - “Figure 5, the authors stated that "The highlighted frequency range below 3GHz is not usable due to small reflection and resonance". Do you have any way to solve this problem?”
Thank you for the question. The resonance and low reflection are related to the tuning element inside the slab coaxial airline. This commercial product is designed for circuit tuning, and we use it beyond its intended application. In circuit tuning, multimode effects in the slab airline are generally not an issue if the desired tuning is achieved.
To improve the reflection behavior for our application, we need to redesign the tuning element, for example, by tapering it. However, manufacturing such a device ourselves is currently beyond our capabilities. - “Section 4.2, could you make a table that compares the advantages and disadvantages of existing propagation constant measurement methods?”
Your feedback is highly appreciated. We added a table comparing our method with other methods (see Table 2).
…In Table 2, we summarize a comparison between the proposed method and existing works in measuring the propagation constant of transmission lines.

Reviewer 3 Report
The manuscript deals with a new method for measuring the propagation constant of the transmission line. The method is both mathematically and experimentally backed. The length of each chapter is adequate. The quality of the presentation is very high. I haven´t found any issues with this manuscript.
Author Response
Thank you for reviewing our paper.